# In-Service Performance Evaluation of Flexible Pavement with Lightweight Cellular Concrete Subbase

Abimbola Grace Oyeyi [1,*], Frank Mi-Way Ni [2] and Susan Tighe [1,3,*]

1   Civil and Environmental Engineering, University of Waterloo, 200 University Ave W, Waterloo, ON N2L 3G1, Canada
2   Department of Civil Engineering, University of Florida, 1949 Stadium Rd., Gainesville, FL 32611, USA
3   Department of Civil Engineering, McMaster University, 1280 Main St W, Hamilton, ON L8S 4L8, Canada
*   Correspondence: a2olaley@uwaterloo.ca (A.G.O.); tighes1@mcmaster.ca (S.T.)

**Abstract:** The objective of engineers to improve the long-term performance of road infrastructure in changing global climate has led to the development of alternate materials for pavement construction. Lightweight cellular concrete (LCC) is a viable option for colder climates where pavements undergo several freeze-thaw cycles each year, resulting in faster deterioration of pavements. This is due to LCCs' excellent freeze-thaw resistance, ease of placement, and potential sustainability benefits such as reduced use of virgin material and industrial by-products. However, there is a need to quantify these benefits and develop unified specifications for using LCC in the pavement structure. Therefore, this study examined the performance of flexible pavement sections that included a subbase layer, unbound granular materials for the control section, and three LCC densities (400, 475, and 600 $kg/m^3$) for the LCC sections. Post-construction evaluation of pavement stiffness and roughness were evaluated using a Lightweight deflectometer and SurPro equipment. The results showed that LCC subbase thickness $\geq$ 250 mm produced over 22% smoother riding surfaces than unbound granular pavements while increasing pavement stiffness by up to 21%. Finally, this study recommends that LCC subbase thickness should not be thinner than 250 mm when using densities below 475 $kg/m^3$ over weak subgrades.

**Keywords:** lightweight cellular concrete; subbase; stiffness; roughness; flexible pavements

## 1. Introduction

Optimizing long-term performance of road infrastructure is a crucial objective for civil engineering, particularly in light of the evolving global climate. In particular, designing and constructing durable pavement structures that balance economic viability and environmental sustainability in regions characterized by low temperatures constitutes a significant challenge. The phenomenon of differential frost heave and the increase in soil moisture content due to prolonged exposure of pavements in cold regions to extremely low temperatures during the winter months, as well as repeated freezing and thawing cycles in the spring, can have detrimental effects on the pavement structure. These effects include the premature development of cracks and a reduction in the bearing capacity of the pavement, leading to shorter service lives and higher maintenance costs [1–4]. As a result, various research studies have been conducted to investigate alternative design and construction techniques, as well as the use of alternative materials. The selection of materials for pavement construction in cold regions should consider factors such as the material's response to changes in climatic conditions, cost, environmental impact, ease of use, and structural capacity. The implementation of innovative materials in the subbase layer of pavement structures has been proposed to address the subgrade weakness issue. Material compositions and functional capabilities of the subbase are deemed imperative to enhance the pavement system's overall structural integrity.

In cold climates, insulation layers are commonly employed directly on the subgrade to mitigate frost penetration within pavements [5]. Various insulation materials have been utilized within the pavement structure to reduce frost penetration, including polystyrene, bottom ash, foamed glass aggregates, foamed concrete, wood residues, and tire chips [6,7]. In addition to their insulating function, several of these materials also serve as a pavement subbase. In contrast, unbound granular layers are not typically regarded as insulating layers due to their thermal conductivity properties. Studies have also shown that these unbound granular material deformation and performance, in general, are influenced by material gradation (size), stress level and moisture content [8,9]. Research has recently demonstrated that lightweight cellular concrete (LCC) possesses superior insulating properties compared to traditional flexible pavements [10,11].

An investigation by Ni [12] evaluated the mechanical properties and suitability of lightweight cellular concrete (LCC) as a subbase material for pavements in Canada. The study determined that LCC possessed greater stiffness than unbound granular materials but a lesser stiffness than chemically stabilized base materials. Additionally, the study identified that the density of LCC plays a crucial role in its properties, with densities of 475 $kg/m^3$ and 600 $kg/m^3$ displaying improved pore structure and durability after 180 cycles of freeze-thaw testing, in contrast to the 400 $kg/m^3$ density LCC. The research suggests that LCC could serve a viable subbase material option, as it offers enhanced durability when compared to traditional materials [12]. At low densities, LCC can exhibit a high percentage of voids ranging from 80–90%, thus minimizing the requirement for raw materials and decreasing the generated waste. Due to the free-flowing nature of LCC, compaction is not required, thus reducing noise pollution and energy consumption associated with compaction [13]. Utilization of industrial by-products, such as slag and fly ash, can reduce waste disposal and increase sustainability [14,15]. The replacement of Portland cement with fly ash, up to a maximum of 75%, in low-density lightweight concrete (LCC) can decrease embodied carbon dioxide ($eCO_2$) while also improving properties such as lower thermal conductivity, reduced dry shrinkage, and diminished heat of hydration [16–19].

## 2. Literature Review

Incorporating a new material in the pavement structure requires evaluation of its performance not only in the laboratory but also in the field to determine its suitability for use. Currently, four types of pavement evaluation are performed by the Ministry of Transportation Ontario (MTO), in Canada, two of which include pavement roughness and pavement structural capacity evaluation [20]. Several factors influence flexible pavement properties, which could serve as an indication of its performance. According to the existing literature, pavement roughness measured as the international roughness index (IRI) (which is an important characteristic of a pavement's longitudinal profile) can be affected by the soil properties, freeze-thaw cycles, moisture content, thickness of the subbase and subgrade layers, and climatic factors [21–24]. Specifically, Lu and Tolliver [25] found that an increase in freeze-thaw cycles results in a roughness increase, and greater roughness deterioration was observed for wet than dry regions.

A study by Von Quintus, Eltahan and Yau [26], concluded that the initial smoothness of new asphalt concrete (AC) pavements strongly influenced roughness advancement, while transverse cracks affected roughness over time for AC and AC overlaid pavements. Pavement roughness can be attributed to more pavement-related parameters such as surface material type, application methods, pavement design, the transition between sections, and pavement deterioration type over time, in terms of frequency, severity, and density [27]. Roughness, in turn, has a substantial impact on fuel consumption, vehicle maintenance costs, and the health of pavement users and their vehicles; therefore, it is a critical pavement measure.

Sayers et al. [28] devised a method to determine approximate ranges of pavement roughness (IRI) in relation to road class and speed features like other studies [29–33]. The influence of pavement roughness is noted to be more significant at higher speeds. As IRI

rises, the speed at which vehicles may safely utilize the roadway falls [27]. While an IRI value of 0 m/km denotes absolute smoothness, an IRI value of 8 m/km denotes a rough, unpaved road surface [34].

In Ontario, Canada, the recommended initial IRI values based on treatment while designing for new or reconstruction AC flexible pavements are 0.8 m/km, 1 m/km, and 1 m/km for freeways, arterial roads, and collector roads, respectively [35]. Local roads have no specified initial IRI; however, the closer to 1 m/km, the better. Similarly, MTO recommends terminal IRI values for a freeway, arterial, collector, and local roadways to be 1.9 m/km, 2.3 m/km, 2.7 m/km, and 3.3 m/km, respectively. However, according to Sayers and Karamihas, [34], newly constructed pavements should range between 1.5 and 3.5 m/km [28]. Automated road profilers like the automated or walking profilers are typically used to determine pavement roughness.

Flexible pavement strength, on the other hand, is influenced by factors such as the quality of materials used and environmental conditions, particularly temperature. The modulus of elasticity (E), also known as stiffness, is commonly used as a measure of strength. A study reported that E increases at lower temperatures and decreases at higher temperatures [36]. The falling weight deflectometer (FWD) is widely used to determine the in-service pavement stiffness [37–39]. The FWD test is used to measure pavement performance by measuring vertical deflection under an impulse load. It uses falling weight and geophone sensors to determine deflections at specific distances from the impact point. The test results are used to estimate the pavement's structural capacity and determine if it's overloaded. The FWD test results are useful for planning future rehabilitation techniques and determining the pavement's capacity and performance life [40]. A portable falling weight deflectometer (also known as a lightweight deflectometer) can perform a similar test as FWD, but it's portable and can be done more often when FWD is unavailable.

Previous research on the application of LCC in pavement structures has primarily focused on laboratory testing of mechanical properties, with limited field performance data and guidelines available. Since pavement roughness and stiffness are influenced by climatic factors and the use of LCC is recommended due to its ability to mitigate some of these influencing factors like freeze-thaw cycling, this study aims to address this gap. This was performed by conducting post-construction evaluations of flexible pavement sections incorporating LCC as a subbase layer, with the objective of quantifying its benefits in terms of pavement, stiffness, and roughness. Thus, performance evaluation and analysis of the LCC pavements would provide insight into the level of serviceability of the pavement, the extent and rate of deterioration over time, and the remaining service life, which could inform decisions related to maintenance requirements.

## 3. Methodology

This study investigated the performance of flexible pavement sections incorporating unbound granular materials as a subbase layer in the control section, and three densities of lightweight concrete (LCC) (400, 475, and 600 kg/m$^3$) in the LCC sections. The in-service evaluation and seasonal impact on pavement stiffness, and roughness were conducted using a lightweight deflectometer and SurPro equipment.

### 3.1. Test Sections

Two field sections in the Region of Waterloo, Ontario, Canada were evaluated. The findings presented in this study are part of an ongoing research that has involved constructing, instrumenting, and monitoring pavements at two locations in the area [11].

The first field section is a 40 m long southbound bus stop lane located on Erbsville Road, Waterloo, Ontario, Canada constructed in October of 2018. This location was selected because yearly maintenance had to be performed at the bus stop location due to severe rutting and cracking. Three segments constitute this section: two 15 m LCC sections and a 10 m control section (Figures 1a,c and 2a–c). Granular B (GB) material with a thickness of 450 mm served as the subbase layer for the control, whereas the LCC sections

used 475 kg/m$^3$ LCC with thicknesses of 350 and 250 mm. Based on the LCC thicknesses, these sections are expressed as LCC350 and LCC250 in this study. The bus stop is located within the LCC350 section. For all sections, a constant 50 mm HL3 and 100 mm HL4 asphalt concrete (AC) surface and base asphalt course were laid over a constant 150 mm of granular A (GA) material. The subgrade on which the pavement sections were constructed was relatively flat. The operating speed for this roadway is 60 km/h.

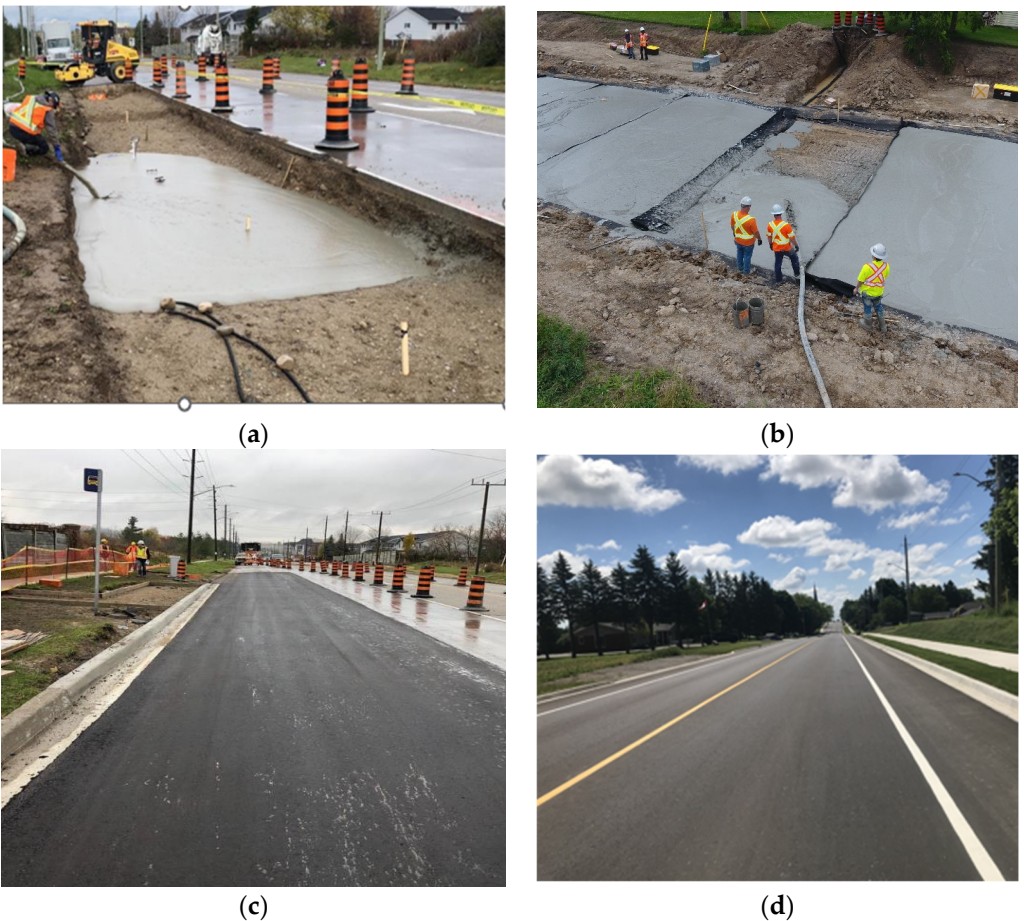

(**a**)  (**b**)

(**c**)  (**d**)

**Figure 1.** Construction and completed Erbsville and Notre Dame Drive (NDD) pavement sections: (**a**) Erbsville LCC pour; (**b**) NDD LCC pour; (**c**) Erbsville completed pavement; (**d**) NDD completed pavement.

The second field section is a 200 m two-lane road located at Notre Dame Drive, St. Agatha, Ontario, constructed in September 2021. This roadway consists of four 50 m segments with a control section having 300 mm GA as base and subbase and three LCC sections with subbases of 200 mm thick 400 kg/m$^3$, 475 kg/m$^3$, and 600 kg/m$^3$ LCC material, respectively (Figure 1b,d and Figure 2d,e). Based on density, the LCC sections will then be referred to as LCC400, LCC475, and LCC600, respectively. Superpave 12.5 AC serves as the surface AC course over Superpave 19.0 AC base course. A 150 mm of GA material was present as the base layer for the LCC sections. The LCC400 and LCC475 sections were constructed over a larger longitudinal slope (1.3%) compared with the other sections. The operating speed for this road is 60 km/h.

The installed temperature and moisture instrumentation layout for the sections at Erbsville and Notre Dame Drive are presented in Figure 3. These sensors were placed in the middle of each pavement layer except moisture sensors that were excluded from the AC layer to measure layer moisture and temperature throughout testing. For the Erbsville and Notre Dame Drive field segments, the subgrade moisture sensor was placed 100 and 150 mm, respectively, within the subgrade. Instrumentation details have been described

in-depth elsewhere [10,11]. Weather stations were also installed at both locations to monitor air temperature and rainfall.

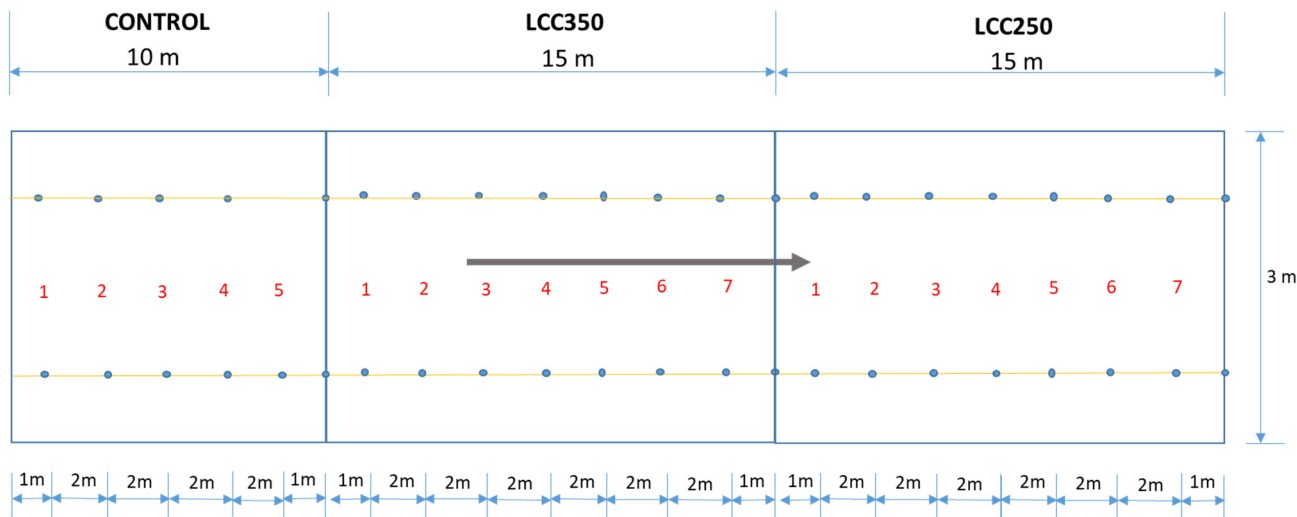

**Figure 2.** Cross section view of pavement sections at Erbsville and Notre Dame Drive (NDD): (**a**) Erbsville control section view; (**b**) Erbsville LCC350 section view; (**c**) Erbsville LCC250 section view; (**d**) NDD control section view; (**e**) NDD LCC400,475,600 typical section view.

**Figure 3.** Lightweight deflectometer testing layout (Erbsville).

### 3.2. Pavement Layer Material Properties

All the material classifications used in this study are common ones that apply in Ontario, Canada. The material properties are presented in Table 1. The testing of materials for AC dynamic modulus (DM) was done in accordance with AASHTO TP 62-07 [41]. The CBR values of the unbound layers were measured in the laboratory in accordance with ASTM D1883 [42], and the values were converted to resilient modulus ($M_R$) using the AASHTOWare Pavement ME method (MEPDG) in Equation (1) [43].

$$M_R = 2555 \times CBR^{0.64} \tag{1}$$

**Table 1.** Notre Dame Drive and Erbsville layer properties (Adopted from [9]).

| Location | Material | Modulus (MPa) | | Poisson's Ratio | |
|---|---|---|---|---|---|
| | | Average | Std Dev | Average | Std. Dev |
| Erbsville | HL3 (21 °C, 5 Hz) | 5716 | 310 | - | - |
| | HL4 (21 °C, 5 Hz) | 5545 | 228 | - | - |
| | Granular A | 226 | 34 | - | - |
| | Granular B | 145 | 37 | - | - |
| | 475 kg/m$^3$ | 1207 | 117 | 0.24 | 0 |
| | Subgrade | 72 | 15 | | |
| Notre Dame | SP 19.0 | 3511 | 87 | - | - |
| | Granular A | 387 | 53 | - | - |
| | 400 kg/m$^3$ | 888 | 107 | 0.21 | 5 |
| | 475 kg/m$^3$ | 1188 | 241 | 0.3 | 14 |
| | 600 kg/m$^3$ | 1391 | 44 | 0.26 | 24 |
| | Subgrade | 43 | 3 | - | - |

Based on ASTM C469 [44], the elastic modulus and Poisson's ratio for the LCC material were calculated. Although the Poisson ratio for the unbound and AC materials was not examined in the lab, typical values for the subgrade type for AC, Granular A, and Granular B in Ontario are 0.35 and between 0.2 and 0.45 (saturated) [35].

Erbsville and Notre Dame granular A met the criteria illustrated by the Ministry of Transportation Ontario [35] and Transportation Association Canada, TAC [45]. According to the unified soil classification system (USCS), the subgrade at Erbsville was determined to be inorganic clays with low or medium plasticity (CL), whereas the subgrade at Notre Dame was determined to be either inorganic clay with high plasticity or organic clays with medium to high plasticity (CL-OH) [46].

### 3.3. Test Methods

#### 3.3.1. Lightweight Deflectometer

The Dynatest portable Lightweight Deflectometer (LWD) was used to determine the pavement stiffness (modulus of elasticity) for the test roads. This test was performed over varying seasons between 2018 and 2022. The LWD has been used because it is portable, less expensive, and readily available, which means it can be used in the absence of the falling weight deflectometer (FWD). However, it is recommended that FWD testing is further performed to validate the LWD results. The test uses a falling weight dropped manually from a constant height onto the pavement surface and deflection readings are obtained and back-calculated to determine the pavement stiffness. The weight applies an impact force of 15 KN and transmits load onto the pavement surface to cover an area of 1.77 m$^2$ using a base plate with a diameter of 150 mm. Before the test, locations were marked for testing (Figures 3 and 4), the testing equipment is shown in Figure 5b. Care was taken to minimize human error by ensuring three consecutive or close outcomes were achieved before moving to the next location. Dropping the load correctly from the required height

also helped with consistency. Equation (2) was used to compute the LWD stiffness based on the elastic half-space (Boussinesq's Solution) theory.

$$E\,(\text{MPa}) = \frac{\pi \times (1 - \mu^2) \times r \times \sigma_0}{2 \times d} \tag{2}$$

where μ is the soil's Poisson ratio, *r* is the plate radius in mm, $\sigma_0$ is the maximum force in kN, and *d* is the maximum deflection in m.

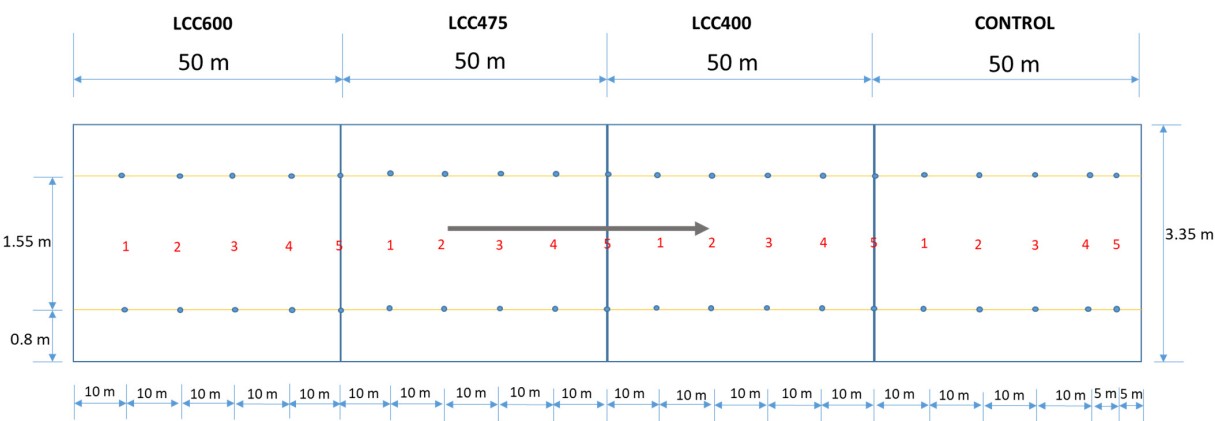

**Figure 4.** Lightweight deflectometer testing layout (Notre Dame Drive).

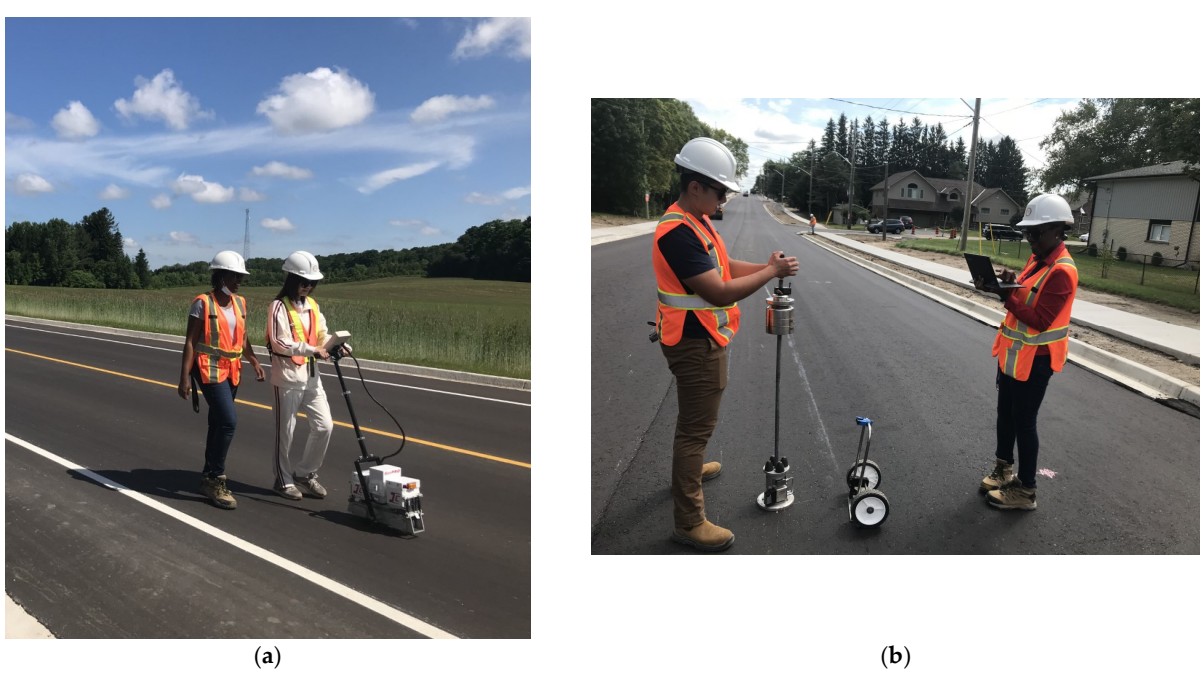

(**a**)                                                        (**b**)

**Figure 5.** Testing equipment during testing at Notre Dame Drive (**a**) lightweight deflectometer (**b**) SurPro.

### 3.3.2. SurPro

The international roughness index (IRI) is a commonly accepted metric for evaluating pavement roughness, calculated by analyzing the longitudinal profile of the road section [47]. Changes in roughness over time, as determined by changes in the longitudinal profile, can serve as a valuable indicator of pavement performance. Typically, profile data are collected at 25 mm intervals using the road profiler. An IRI value of 0 m/km represents absolute smoothness, while 10 m/km represents the roughness of an unpaved road [47]. Pavement roughness was quantified utilizing the SurPro walking profiler.

The SurPro is a laser walking profiler which uses a rolling inclinometer to measure the true unfiltered profile of the pavement and calculate the IRI. The test was performed by walking along the right wheel paths (RWP) and left wheel path (LWP) following the traffic direction, two times at least on each wheel path, at a speed no more than 25 m/s (Figure 5a). Inclinometers in the profiler measure relative changes in elevation between the wheels, which converts to IRI for assessing the roughness. This test was performed on the same days as the LWD tests.

## 4. Results

### *4.1. Pavement Stiffness*

The LWD test was used to evaluate the LCC pavement stiffness over time. Testing was performed at different months to estimate the seasonal effect on the stiffness of the alternative designs. Test spots at each location were predetermined. As a result, the LWD tests were performed at roughly the same location each time, with a tolerance range of 1 m². This section presents and discusses the results for Erbsville and Notre Dame Drive.

#### 4.1.1. Erbsville

Figure 6 presents the elastic modulus and the mean stiffness (average of the RWP and LWP) for each testing season, while Figure 7 presents the deflection results for all the testing periods at Erbsville. The error bars represent one standard deviation from the mean in the positive and negative directions. A *t*-test was performed to confirm whether the RWP and LWP had different stiffness. The findings suggested that, at a 95% confidence level, there was no significant difference in February and April of 2022 (*p*-value 0.92 and 0.97, respectively). On the contrary, a considerable difference was found between these wheel paths in June 2022 (*p*-value 0.02).

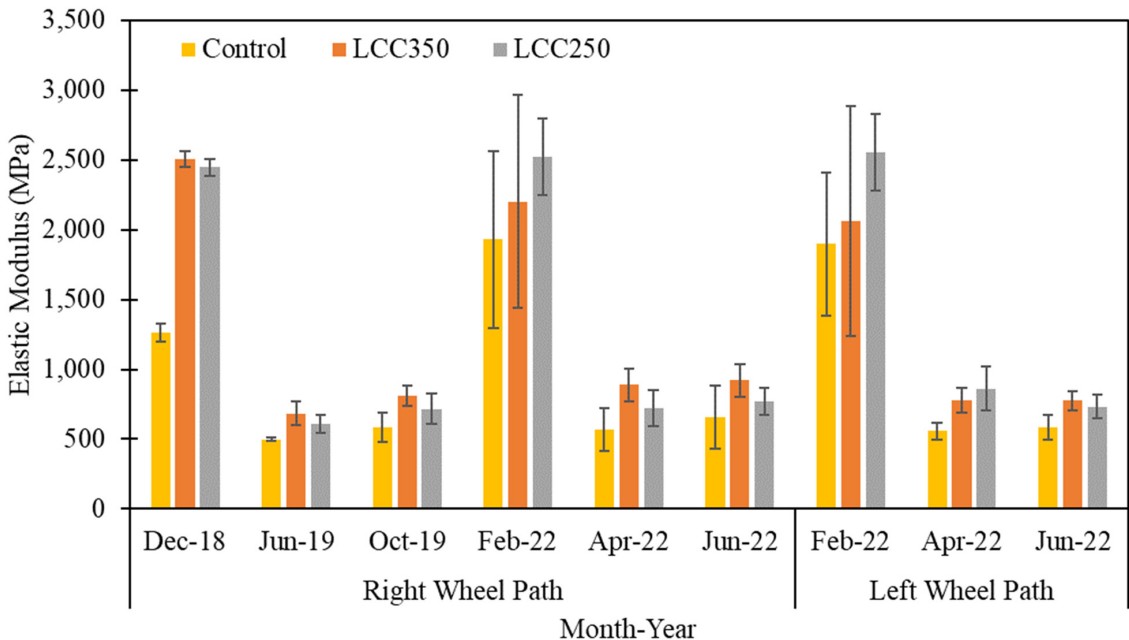

| Section | Section Mean Elastic Modulus (MPa) | | | | | |
|---|---|---|---|---|---|---|
| | **Dec-18** | **Jun-19** | **Oct-19** | **Feb-22** | **Apr-22** | **Jun-22** |
| Control | 1265 | 498 | 585 | 1914 | 562 | 621 |
| LCC350 | 2506 | 684 | 811 | 2132 | 834 | 849 |
| LCC250 | 2449 | 610 | 717 | 2538 | 790 | 751 |

**Figure 6.** Erbsville LWD measured elastic modulus.

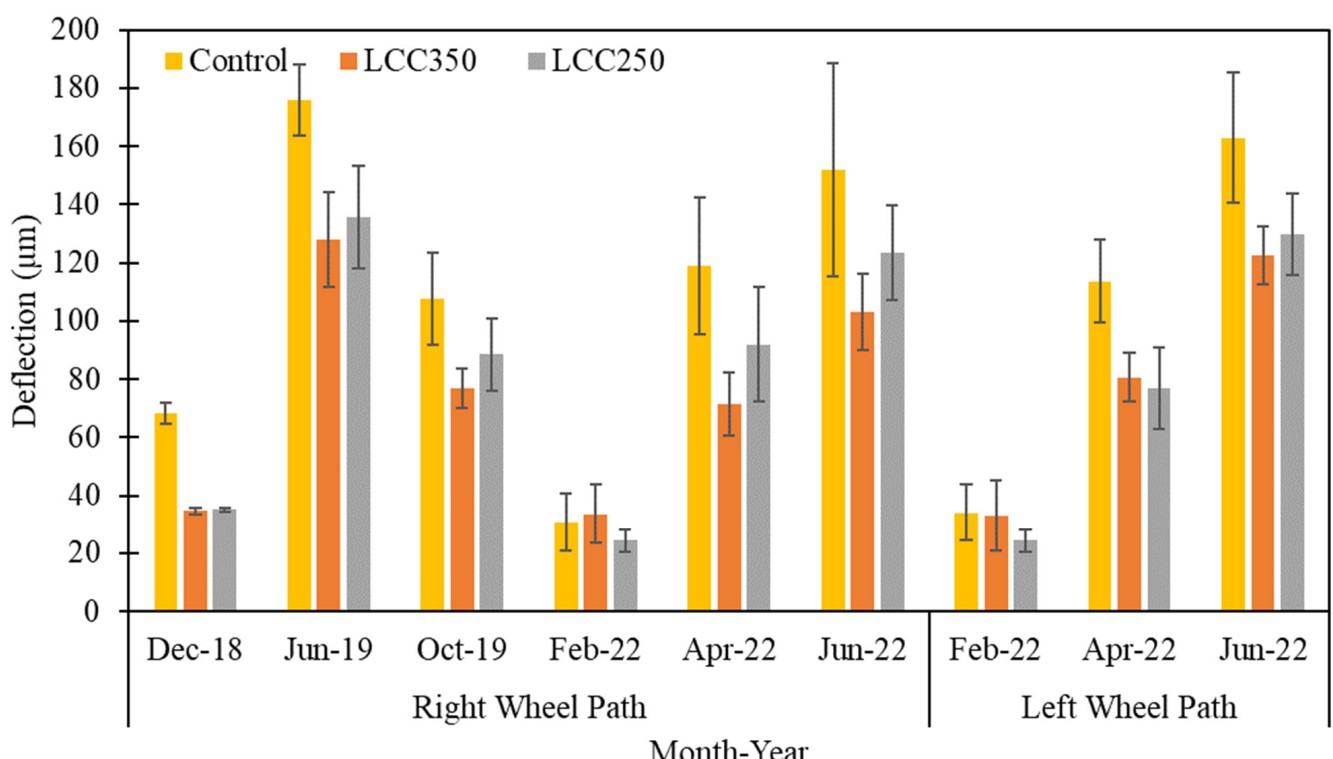

**Figure 7.** Erbsville LWD measured deflection.

The initial measured pavement elastic modulus (indicative of the pavement stiffness) was observed to be the most in the LCC350 (2506 MPa), followed closely by the LCC250 section (2449 MPa) and then the control (1265 MPa). However, these values only represent near freezing or freezing conditions, producing stiffer pavement layer conditions. In addition, only two testing points were completed per section at this time, which could be the reason for the wide gap in stiffness between the LCC and control sections. As can be noted with the error bars (and within each section), some variability in stiffness was seen. A similar trend was seen over the analysis period, with the LCC350 section having the stiffest pavement structure and the control having the least. Additionally, lower deflection values produced higher stiffness, as expected. A study by Hossain and Apeagyei, [48] also found spatial variability in pavement subbase and subgrade stiffness when LWD and two other testing equipment were used to assess the pavement stiffness. However, this assessed unbound layers and inferred that the variability could be attributed to soil suction or pore pressure development due to transient loading of the LWD on fine-grained soil.

Stiffness growth occurred over time. Comparing June 2019 and June 2022, a stiffness increase of 25% was found for the control, 24% for LCC350, and 23% for LCC250. The slightly lower increase for the LCC sections compared to the control could be because of these sections' subjection to continuous bus and turning traffic over time. Typically, the bus stops within these two sections, and most of the traffic turning right at the intersection stop and start at the LCC250 section. This could also explain why between December 2018 and February 2022, a stiffness decrease (15%) was observed only in the LCC350 section. Other factors, such as moisture content and layer temperature, could influence pavement stiffness [48].

The significant difference between the measured stiffness for the sections at a 95% confidence level was tested with the least square difference (LSD) analysis. A significant difference existed between the control and LCC sections, especially in December 2018 and June 2019. However, in October 2019, the control displayed stiffness not significantly different from LCC250. In addition, in 2022, the results showed no significant difference in measured stiffness for all the sections. Only measured stiffness values on the LWP for the

LCC sections in April and June 2022 were significantly different from that of the control. Again, since the LCC sections are the most subjected to vehicular traffic, it is expected that deterioration should occur more rapidly than the control; this could be why the stiffness values were closer to that of the control in later years. That being said, the LCC sections still exhibited higher stiffness than the control.

In conclusion, because only the subbase of the pavement structure was altered, it may be argued that the LCC layer enhanced the pavement structure's stiffness compared to the unbound granular B. The stiffness improvement was proportional to the thickness of the LCC.

LSD analysis was done on the measured stiffness values between each testing month to determine if there was a seasonal effect on pavement stiffness. Seasonal stiffness analysis showed that the period when the stiffness was performed was a significant factor in determining its magnitude. The 95% LSD criteria were obtained to be 589 MPa for the control, 504 MPa for LCC350, and 191 MPa for LCC250. No significant difference was recorded in all the sections (at a 95% confidence level) for stiffness measured between April, June, and October when compared with each other. The relationships between these months had mean differences over two times below the 95% LSD criteria. However, when comparing the stiffness between other months such as December/April, December/June, and December/October, the mean difference in stiffness was found to be one time higher for the control, over three times greater for the LCC350 section, and more than nine times greater for LCC250 section than the LSD criteria. Similar outcomes were achieved when February stiffness was compared with April, June, and October stiffnesses. A comparison between February and December indicated no difference for the LCC section but noted a significant difference within the control. This could further show the insulating properties of the LCC section during the early cold period yielding some difference.

The LWD results were found to be consistent with FWD testing results for these road sections. Although the magnitude of the elastic modulus and deflections were higher, due to greater pavement coverage by the FWD equipment, the FWD results showed greater stiffness for the LCC sections than the control, and the LCC350 results were greater than those of the LCC250 section [49].

### 4.1.2. Notre Dame Drive

The *t*-test result for Notre Dame Drive indicated that there was no significant difference between the stiffness measured on the RWP and LWP for September 2021 and February, April, and June 2022 (*p*-values of 0.34, 0.66, 0.09, and 0.10, respectively) at 95% confidence level. Figure 8 shows the mean measured stiffness on each wheel path on the southbound lane and the section (RWP and LWP) mean elastic modulus, while Figure 9 presents the measured deflection on each wheel path. The error bars represent one standard deviation from the mean in the positive and negative directions.

In September 2021, the initial elastic modulus for all sections was discovered to be relatively low compared with typical flexible pavements. The stiffness of the control was 224 MPa, followed by 187 MPa, 146 MPa, and 96 MPa for the LCC600, LCC475, and LCC400, respectively. Because the testing was done a few hours after AC was installed, the low stiffness could be attributed to the softer asphalt concrete layer at the time. During testing, the surface layer temperatures were 47 °C, 45 °C, 43 °C, and 47 °C for the control, LCC400, LCC475, and LCC600, respectively. The temperatures of the base, subbase, and subgrade were also found to be between 25 °C and 20 °C, with the LCC base layers averaging 5 °C higher than the control's temperature and the subbase and subgrade averaging 2 °C lower.

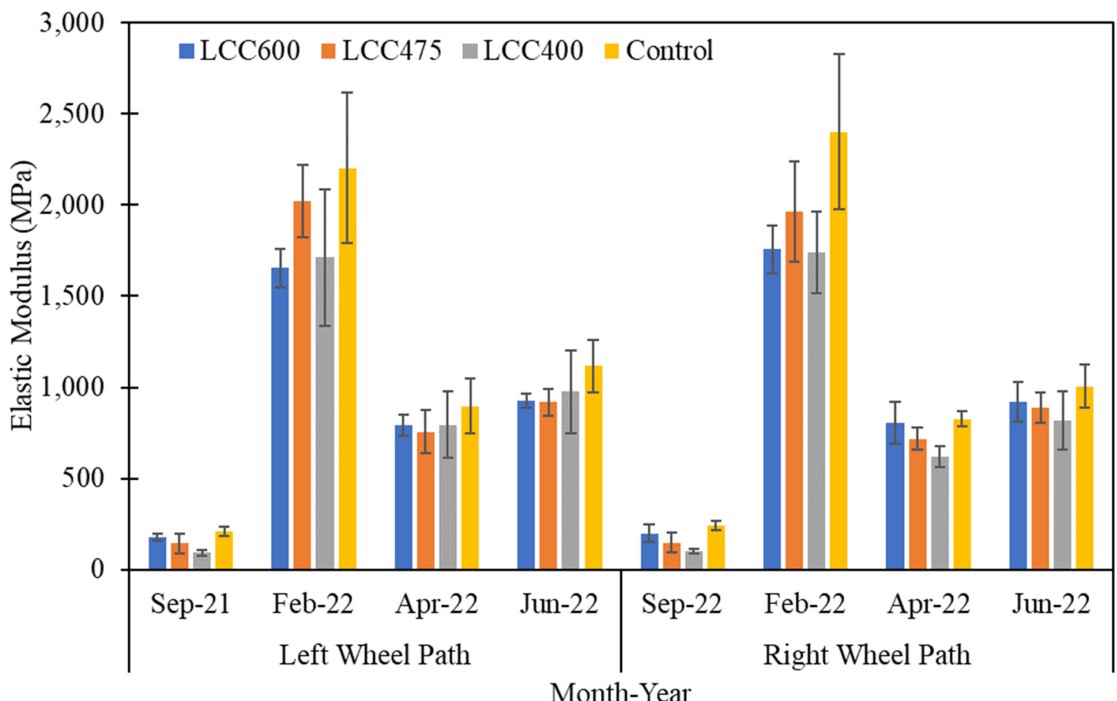

| Section | Section Mean Elastic Modulus (MPa) | | | |
|---------|--------|--------|--------|--------|
|         | **Sep-21** | **Feb-22** | **Apr-22** | **Jun-22** |
| LCC600  | 187  | 1706 | 799 | 922  |
| LCC475  | 146  | 1993 | 737 | 903  |
| LCC400  | 96   | 1726 | 706 | 898  |
| Control | 224  | 2302 | 861 | 1061 |

**Figure 8.** Notre Dame Drive southbound measured elastic modulus.

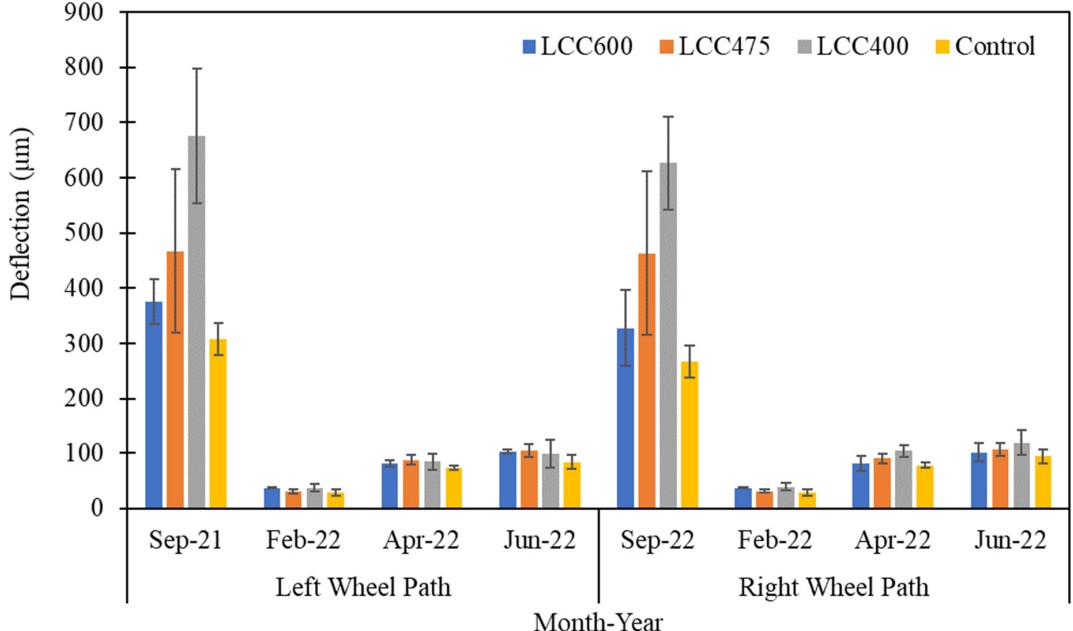

**Figure 9.** Notre Dame Drive southbound measured deflection.

During the testing period, the control portion was stiffer than the LCC sections, contrary to the results observed at Erbsville. Construction method, insufficient compaction (no vibratory compaction done during construction) on the LCC sections, excessive truck traffic before AC placement, rainfall before pour softening subgrade, and the fact that the LCC475 section and a significant portion of the LCC400 were placed on a longitudinal slope greater than 1.3 percent are thought to be factors that contributed to the lower stiffness values. Because LCC is a stiffer material, it is typically projected to provide a stiffer pavement structure, like Erbsvilles' [50]. However, the lower thickness could also be an influencing factor. It is recommended that the pavement sections be monitored to see if improvements in the LCC sections continue as expected.

Comparing stiffness between September 2021 and April 2022, the disparity in stiffness between the control and LCC sections decreased by 86%, 68%, and 61% for LCC400, LCC475, and LCC600, respectively. This could signify that the compaction level within the sections had increased over time due to traffic, and the pavement is gaining strength and stability to support the pavement structure adequately. In addition, the stiffness of the LCC sections was proportional to their density.

Furthermore, the LSD method was applied to assess if there was a difference in measured stiffness between sections (Table 2). The ANOVA analysis first indicated that, between September 2021 and April 2022, there was a significant difference between some sections, with *p*-values below 0.05 at a 95% confidence level. In Table 2 the highlighted portion in boxes indicates the areas with a significant difference in measured stiffness. A significant difference was noted between LCC400 with LCC600 and control initially. However, after the final AC paving in June 2022, the measured stiffness indicated no statistical difference between the sections. This could further show improvement in the stiffness of the LCC sections over time.

**Table 2.** Notre Dame Drive mean least square difference (LSD) analysis.

| Section Relationship | Section Mean Difference | | | | | | | |
|---|---|---|---|---|---|---|---|---|
| | Sep-21 | | Feb-22 | | Apr-22 | | Jun-22 | |
| | RWP | LWP | RWP | LWP | RWP | LWP | RWP | LWP |
| Control-LCC400 | 140 | 109 | 663 | 490 | 209 | 103 | 186 | 141 |
| Control-LCC475 | 92 | 58 | 441 | 179 | 108 | 142 | 117 | 199 |
| Control-LCC600 | 42 | 26 | 645 | 548 | 20 | 105 | 88 | 191 |
| LCC400-LCC475 | 48 | 51 | 222 | 311 | 101 | 39 | 69 | 59 |
| LCC400-LCC600 | 98 | 83 | 18 | 58 | 189 | 2 | 98 | 50 |
| LCC475-LCC600 | 50 | 32 | 204 | 369 | 88 | 37 | 29 | 8 |
| **LSD Criteria** | 70 | 54 | 504 | 563 | 133 | - | - | - |
| ***p*-value** | 0.000 | 0.000 | 0.007 | 0.041 | 0.002 | 0.415 | 0.153 | 0.134 |

The stiffness immediately after the base asphalt was placed in September of 2021 showed a smoother profile along the road section compared with after placement of the surface asphalt concrete layer, where more variability in stiffness within each section was seen. The most variability was observed in the control section, then the LCC400 section. The LCC600 section showed the most constant stiffness along its length. As expected, the pavements' elastic modulus on average post-final AC paving increased by 5, 6, 11, and 5 times for the control, LCC400, LCC475, and LCC600 sections, respectively. The biggest stiffness jump was noted in the LCC400 section.

In assessing seasonal effects on each section, LSD results showed that relationships between all the months except April and June were significantly different. This further proves that temperature and moisture are the main determinants of pavement stiffness, as indicated in other studies [36,51]. A change in the asphalt concrete surface temperature, which impacts stiffness, could alter the stress condition across the pavement. Due to their

typical stress reliance, the underlying unbound layers' performance may be affected by this shift in stress state [51].

Regression analysis was performed between surface temperature for all the sections excluding LCC600 with the respective measured elastic modulus. The result showed a strong relationship ($R^2 > 0.8$) in all the sections (Figure 10). When regression was performed with the base, subbase, and subgrade layers, the relationship was less strong and decreased with depth. The ambient temperature over the study period was 22 °C, −2 °C, 12 °C, and 24 °C for September 2021, February, April, and June 2022, respectively.

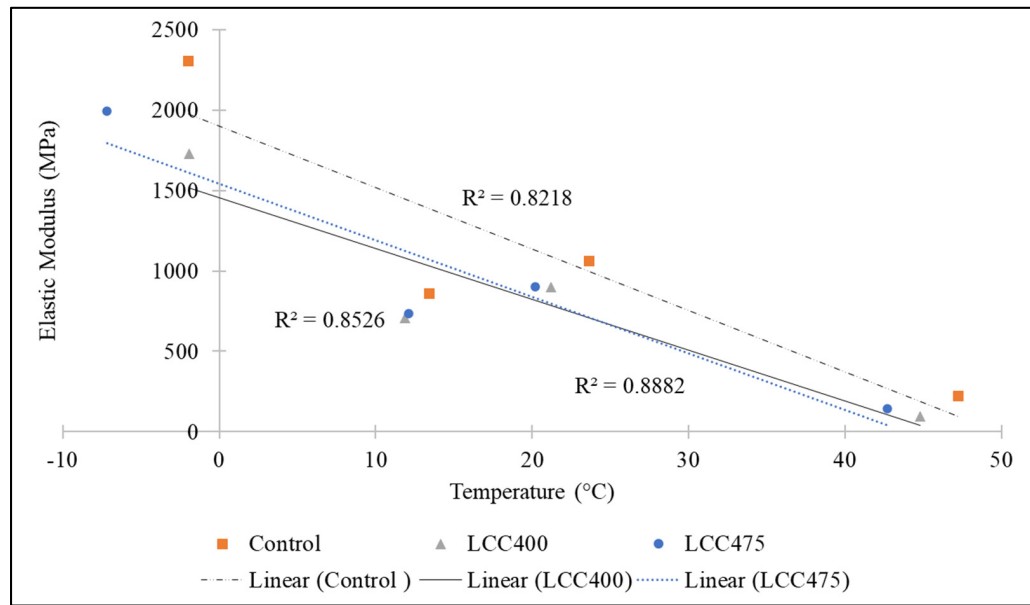

**Figure 10.** Stiffness relationship with pavement surface layer temperature.

### 4.2. Pavement Roughness

#### 4.2.1. Erbsville

The average IRI on each wheel path for the SurPro was calculated using the readings from all the runs. The findings of the ANOVA analysis showed that there was statistically no distinction between each run in a wheel path. For instance, all the sections for December 2018 and June 2019 had *p*-values above 0.6 at a 95% confidence level. The mean IRI on each wheel path and mean section IRI values over the study period are presented in Figure 11. Figure 12 presents the coefficient of variation from the mean for measured IRI. The error bars in the figure represent the standard deviation for each location.

Erbsville's control section consistently had higher IRI than the LCC sections on both wheel paths. For LCC sections, the LCC350 at the earlier months for both wheel paths had a higher IRI, but for some of the tests, especially in 2022, the LCC250 IRI was greater on both wheel paths. There was higher variability in pavement roughness on the RWP than on the LWP. IRI also decreased from 2018 to 2019. IRI was observed to fluctuate depending on the testing time. The results also reflect that a large portion of the Erbsville control section measured IRI was above 4 m/km in June 2019 and 2022. The control had 44% more points above 4 m/km than LCC350 and 9% more points than LCC250, even though these sections were 50% longer than the control. The highest IRI for the LCC350 section was focused within 15 to 20 m of its length. This was the location of the bus stop; this explains why those areas have IRI above 4 m/km. Notwithstanding, the bus stop location has not required any maintenance or rehabilitation as in the past and is in good condition over three years. At the LCC250 section, the highest IRI was noted at the edge of the section, which was the location for left-turning traffic.

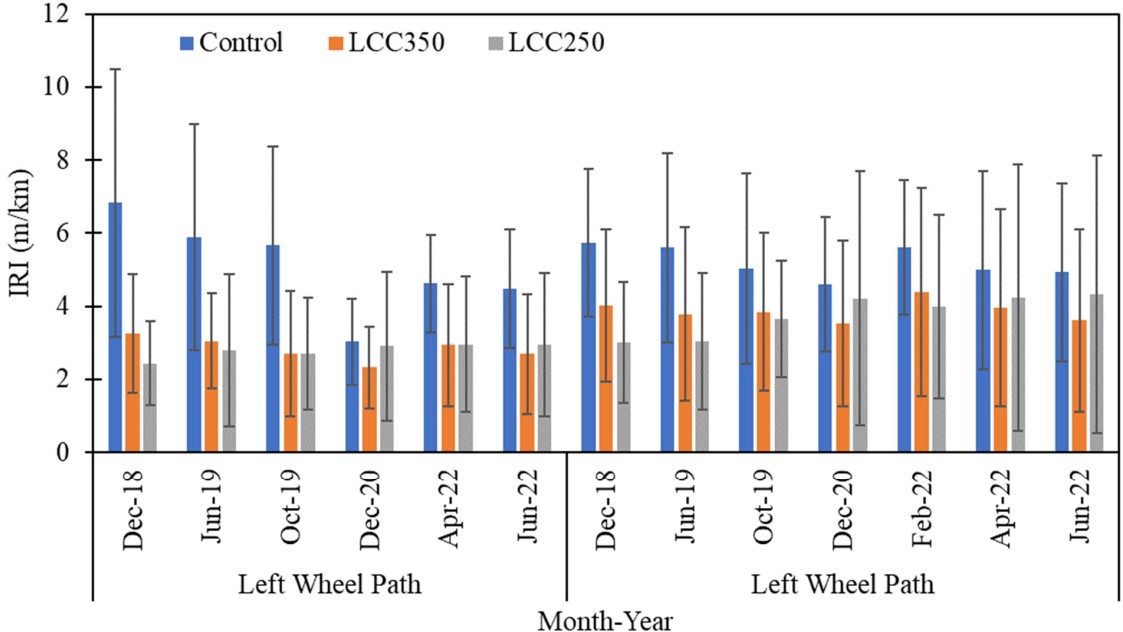

| | **Section Mean IRI (m/km)** | | | | | | |
|---|---|---|---|---|---|---|---|
| | **Dec-18** | **Jun-19** | **Oct-19** | **Dec-20** | **Feb-22** | **Apr-22** | **Jun-22** |
| Control | 6.27 | 5.74 | 5.35 | 3.81 | 5.12 | 4.99 | 4.70 |
| LCC350 | 3.64 | 3.41 | 3.27 | 2.93 | 3.66 | 3.96 | 3.15 |
| LCC250 | 2.72 | 2.91 | 3.18 | 3.56 | 3.47 | 4.23 | 3.63 |

**Figure 11.** Erbsville measured IRI.

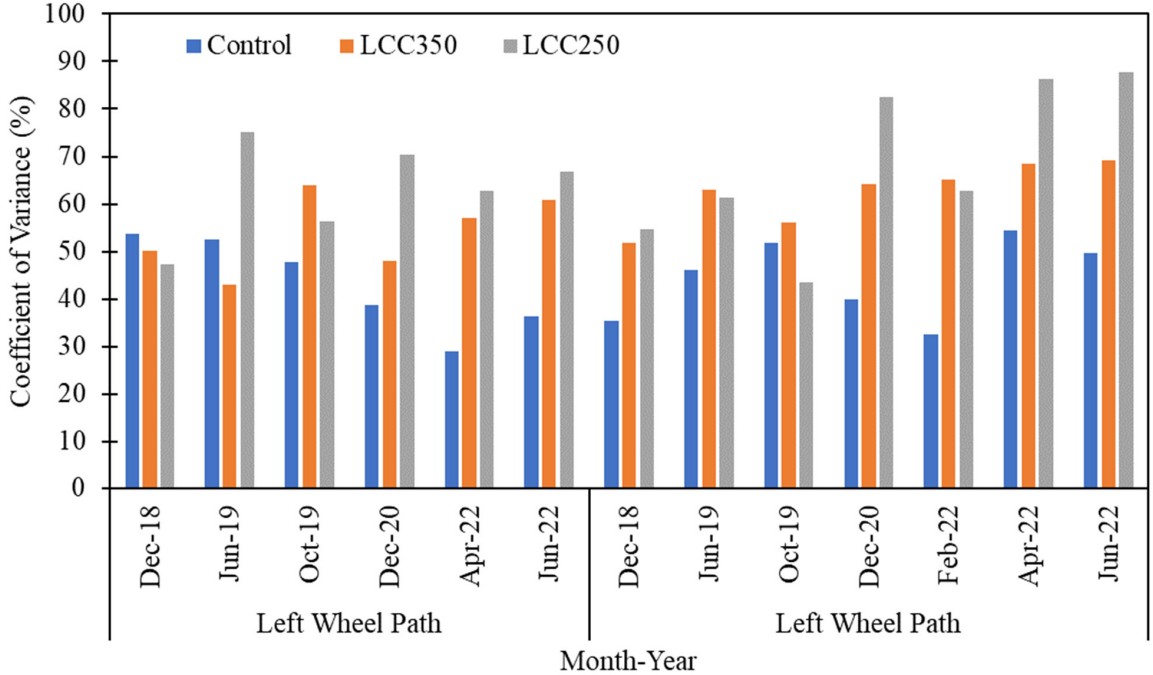

**Figure 12.** Erbsville measured IRI coefficient of variation from the mean.

The LSD result revealed that for the first few years, the LCC portions performed noticeably differently from the control section, but by 2022, there was no difference between all the sections. Due to the impact of bus and vehicle traffic on the LCC portions compared to the control section, the IRI gap between sections may have been closer than in previous years. By June 2022, the control IRI at Erbsville was an average of 49% more than the LCC350 section and 43% more than the LCC250 section.

### 4.2.2. Notre Dame Drive

The IRI results for the southbound RWP and LWP are presented in Figure 13 and the coefficient of variation from the mean in Figure 14. The mean IRI values for each direction are provided in Table 3. The error bar represents one standard deviation from the mean.

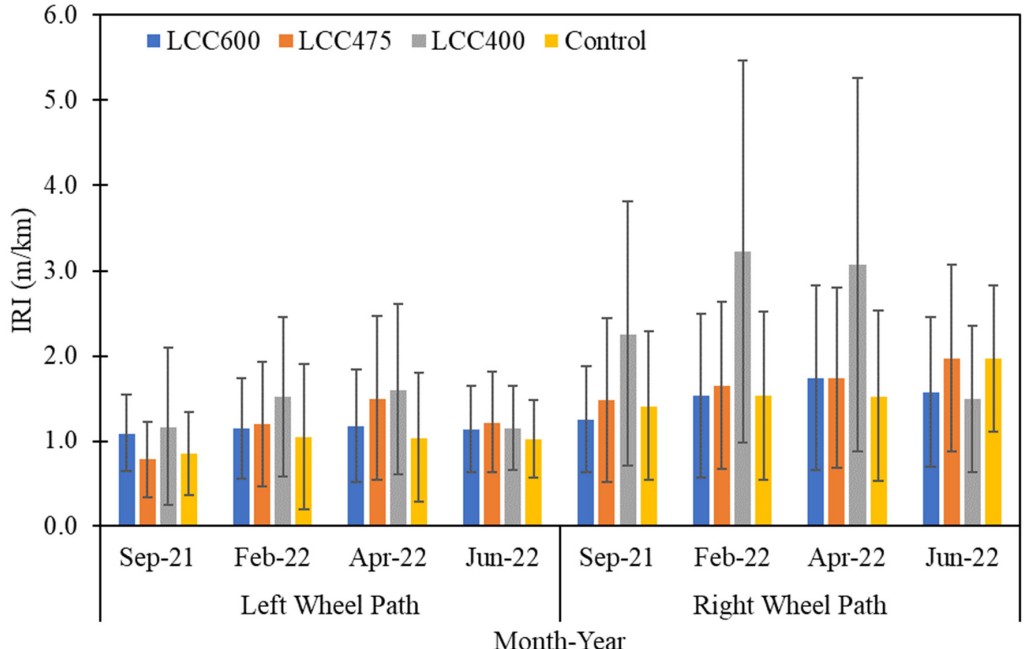

**Figure 13.** Notre Dame Drive Southbound Lane measured IRI.

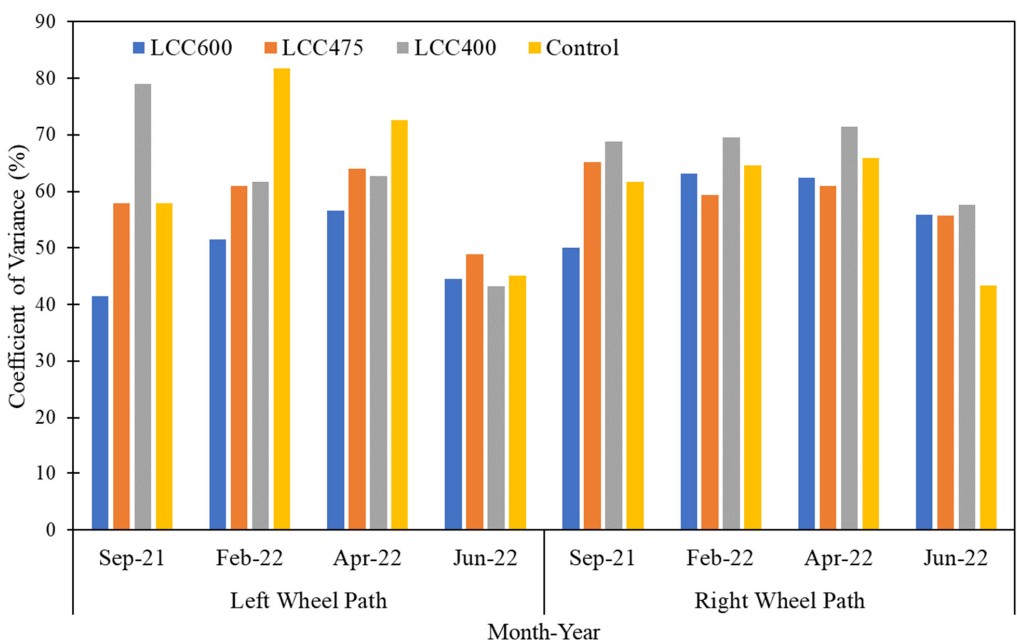

**Figure 14.** Notre Dame Drive Southbound Lane measured Coefficient of Variance.

**Table 3.** Notre Dame Drive mean IRI.

| Section | Southbound Section Mean IRI (m/km) | | | |
|---|---|---|---|---|
| | Sept 21 | Feb 22 | April 22 | June 22 (Baseline IRI) |
| LCC600 | 1.17 | 1.34 | 1.46 | 1.36 |
| LCC475 | 1.13 | 1.43 | 1.62 | 1.60 |
| LCC400 | 1.71 | 2.37 | 2.34 | 1.33 |
| Control | 1.13 | 1.29 | 1.28 | 1.50 |
| **Northbound Section Mean IRI (m/km)** | | | | |
| LCC600 | 1.23 | | 2.00 | 1.28 |
| LCC475 | 1.26 | | 1.78 | 1.23 |
| LCC400 | 1.50 | | 1.50 | 1.82 |
| Control | 1.33 | | 1.18 | 1.68 |

Before the final lift of asphalt was placed in June 2022, on both wheel paths the most IRI occurred in April 2022. There was some variation in the IRI measurements along the section profiles, as shown by the error bars. When comparing June 2021 and June 2022, which are representative of the trend for all the other testing times, it was observed that the control and LCC600 sections performed similarly with data points mostly below 2 m/km. However, LCC475 and LCC400 had a lot of points above 2 m/km. This could be because these locations were situated on slopes greater than 1.3 percent, which could have resulted in lower thicknesses at these locations. Additionally, before final surface paving, the LCC400 section experienced depressions along its length, which could have contributed to the results. The depressions coincided with former areas of potholes and could also be because of inadequate compaction and the disturbance of the LCC layer due to heavy rainfall a few hours after it was placed. Judging by the 2022 IRI measurements, the LCC400 section appeared to have performed better than prior testing.

After final asphalt lift placement, more consistent IRI was achieved for all the sections on all wheel paths except the northbound RWP, where the LCC400 and control section experienced the most IRI (Table 3). Overall, by June 22, the control exhibited an average of 10% and 13% more roughness than the LCC600 and LCC400 sections and 7% less roughness than the LCC475 on the southbound lane. On the northbound lane, the control had a roughness of 31% and 37% more than LCC600 and LCC475, while the LCC400 had the greatest roughness, 8% more than the control section. LSD analysis revealed that the LCC sections and control performed differently at various times on the northbound and southbound lanes.

## 5. Conclusions

- The field experiment was carried out to investigate the in-service pavement stiffness and roughness of lightweight cellular concrete subbase pavements and compare them with traditional unbound materials subbase pavements. This was done by examining the deflections induced by a lightweight deflectometer to measure the pavement stiffness and a Surpro walking profiler to measure the pavement roughness over varying seasons. Based on these evaluations, the following conclusions have been made.

- The findings show that flexible pavements with LCC subbase thicknesses $\geq$ 250 mm could produce a 21% stiffer pavement structure compared with twice as thick unbound granular B subbase pavements. An increase of 36% in stiffness can occur when LCC thickness increases by 100 mm. Pavement stiffness was also noted to increase with LCC layer density and over time. LCC subbase thickness $\geq$ 250 mm can produce over 22% smoother riding surfaces than unbound granular pavements. Time of testing, pavement temperature during testing, construction method, insufficient compaction, excessive truck traffic on LCC pavements before AC placement, and road gradient could influence LCC pavement stiffness.

- LCC thickness equal to 200 mm could yield over 20% smoother pavements for a 600 kg/m$^3$ LCC subbase. While varying results were seen for 400 kg/m$^3$ and 475 kg/m$^3$ LCC subbase pavements compared to unbound subbase materials due to varying reasons discussed, it was evident that they may function comparably or better than conventional pavements. Factors influencing roughness progression in LCC pavements include environmental conditions, seasonal variations, road class, road function, road gradient, subgrade conditions, initial construction process, and practices (compaction, early vehicular traffic).
- It is recommended that pavement sections be monitored over a longer period and that a correlation between LWD and FWD testing be performed to further assess the pavement sections' performance.

**Author Contributions:** Conceptualization, A.G.O., F.M.-W.N. and S.T.; methodology, A.G.O. and F.M.-W.N.; software, A.G.O.; formal analysis, A.G.O.; data collection, A.G.O. and F.M.-W.N.; data curation, A.G.O.; writing—original draft preparation, A.G.O.; writing—review and editing, A.G.O., F.M.-W.N. and S.T.; supervision, S.T.; funding acquisition, S.T. All authors have read and agreed to the published version of the manuscript.

**Funding:** This research was funded by Natural Sciences and Engineering Research Council of Canada (NSERC) and CEMATRIX, grant number funding reference CRDPJ 514908–2017.

**Institutional Review Board Statement:** Not applicable.

**Informed Consent Statement:** Not applicable.

**Data Availability Statement:** Not applicable.

**Acknowledgments:** The authors of this research gratefully acknowledge CEMATRIX (CANADA) Inc., the Region of Waterloo, Ontario, Canada, Natural Sciences and Engineering Research Council of Canada (NSERC), and Centre for Pavement and Transportation Technology (CPATT), University of Waterloo for supporting this research.

**Conflicts of Interest:** The authors declare no conflict of interest.

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
