# Peer review of "In-Service Performance Evaluation of Flexible Pavement with Lightweight Cellular Concrete Subbase"

_applsci, doi:10.3390/app13084757_

Round 1

Reviewer 1 Report

Dear Authors, 

1. The paper has several typos. Authors need to proofread the paper to eliminate all of them.

2. The text of some figure(s) is too small. Authors should make sure that the text can be read if printed on paper.

3. In introduction avoid having figure

4. The paper should be updated to include more recent references, preferably from the last 3 or 4 years.

5. The authors should add a clear and detailed problem definition. 

6. There is not enough discussion of the test section and results. 

7. Some text must be added to discuss the future work or research

Author Response

Dear Reviewer,
Thank you very much for all the comments and suggestions. They were very helpful in updating this work and making sure that our work is well understood by the readers. We really appreciate the detail and effort put into reviewing this work.

We have provided some clarifications for each raised concern or comment (in bold) below and hope this is adequate in addressing them. We are happy to provide further clarification should this be needed.

  1. The paper has several typos. Authors need to proofread the paper to eliminate all of them. (Paper has been proofread)
  2. The text of some figure(s) is too small. Authors should make sure that the text can be read if printed on paper (Figures 2, 3, and 4 text have been increased).
  3. In the introduction avoid having figure (Figure removed)
  4. The paper should be updated to include more recent references, preferably from the last 3 or 4 years. Some new references like Jiang, 2022, Hettiarachchi, 2023 and Stehlik 2022)
  5. The authors should add a clear and detailed problem definition. (This has been included in the last paragraph of section 2)
  6. There is not enough discussion of the test section and results (Additional text and pictures during testing have been added)
  7. Some text must be added to discuss the future work or research (This has been added in the last paragraph)

Reviewer 2 Report

This is a carefully done study and the findings are of considerable interest. A few minor revisions are list below.

1           The introduction can be split into two sections: introductions and literature background.

2           In the introduction section, it is recommended that some references that focus on mechanical property of unbound granular materials should be added, including but not limited to:

Ullah S, Jamal A, Almoshaogeh M, et al. Investigation of Resilience Characteristics of Unbound Granular Materials for Sustainable Pavements[J]. Sustainability, 2022, 14(11): 6874.

Li J, Zhang J, Yang X, et al. Monte Carlo simulations of deformation behaviour of unbound granular materials based on a real aggregate library[J]. International Journal of Pavement Engineering, 2023, 24(1): 2165650.

Stehlik D, Hyzl P, Dasek O, et al. Comparison of Unbound Granular Materials’ Resilient Moduli Determined by Cyclic Triaxial Test and Innovative FWD Device[J]. Applied Sciences, 2022, 12(11): 5673.

3           Testing methods are an interesting component to the work, but are not outlined in sufficient detail.

4           Please revise the conclusion in paragraphs. Conclusions are not just about summarizing the key results of the study, it should highlight the insights and the applicability of your findings/results for further work. Please make it more concise and show only the high impact outcomes.

Author Response

Dear Reviewer,
Thank you very much for all the comments and suggestions. They were very helpful in updating this work and making sure that our work is well understood by the readers. We really appreciate the detail and effort put into reviewing this work.

We have provided some clarifications for each raised concern or comment (in bold) below and hope this is adequate in addressing them. We are happy to provide further clarification should this be needed.

  1. The introduction can be split into two sections: introductions and literature background. (The introduction has been split into two sections)
  2. In the introduction section, it is recommended that some references that focus on mechanical property of unbound granular materials should be added, including but not limited to: (some discussion on the performance of unbound material has been added to the introduction and referenced accordingly, however, since the focus of this study is LCC we have limited this discussion)

Ullah S, Jamal A, Almoshaogeh M, et al. Investigation of Resilience Characteristics of Unbound Granular Materials for Sustainable Pavements[J]. Sustainability, 2022, 14(11): 6874.

Li J, Zhang J, Yang X, et al. Monte Carlo simulations of deformation behaviour of unbound granular materials based on a real aggregate library[J]. International Journal of Pavement Engineering, 2023, 24(1): 2165650.

Stehlik D, Hyzl P, Dasek O, et al. Comparison of Unbound Granular Materials’ Resilient Moduli Determined by Cyclic Triaxial Test and Innovative FWD Device[J]. Applied Sciences, 2022, 12(11): 5673.

  1. Testing methods are an interesting component to the work, but are not outlined in sufficient detail. (Additional text and pictures during testing have been added)
  2. Please revise the conclusion in paragraphs. Conclusions are not just about summarizing the key results of the study, it should highlight the insights and the applicability of your findings/results for further work. Please make it more concise and show only the high impact outcomes.(conclusions have been edited and summarized)

Reviewer 3 Report

11.       Abstract: The two sentences “The results showed that … up to 21%” and “Thus, it is recommended … weak subgrades” have no strong logic relationship with each other. The prior sentence indicates the benefits of thickness 250 mm, while this is not the complete reason that the thickness <250 mm “should not” be used. More explanation is required between the two sentences.

22.       Section 1: The sentence “Since pavement roughness and stiffness are influenced by climatic factors and the use of LCC is recommended due to its ability to mitigate some of these influencing factors like freeze-thaw cycling this study aims to address this gap by conducting post-construction evaluations of flexible pavement sections incorporating LCC as a subbase layer, with the objective of quantifying its benefits in terms of pavement, stiffness, and roughness” should be divided into two sentences.

33.       Section 3.1: The full names of AC, GA, and GB should also be explained in the title of Figure 3, so that readers would not check them in the main text. In addition, why there are two “(b) Erbsville LCC350 section view” in the title of Figure 3?

44.       The section numbers are not in sequence. For example, the subsection number of Section 2 is 3.1 (should be 2.1), two “section 3.1” in the paper, etc. The authors should be serious to these details.

55.       It is not very necessary to present Table 2 separately. The contents can be included in Figure 6 next to RWP and LWP. Same for Table 3 and Figure 8, Table 5 and Figure 11, Table 6 and Figure 12

66.       Table 4: Other marks instead of red color should be used to highlight the results, considering the formatting and printing of the journal.

77.       Figure 10: Axes names are missing.

88.       Since LCC is advantageous for low carbon footprint, a short section with primary estimation of CO2-eq is required to complement the paper, considering the different raw materials production, construction, and service life between LCC and the control subbase.

99.       Conclusion: There is a lack of outlook to figure out the future research based on the results (or limitations) of this paper.

Author Response

Dear Reviewer,
Thank you very much for all the comments and suggestions. They were very helpful in updating this work and making sure that our work is well understood by the readers. We really appreciate the detail and effort put into reviewing this work.

We have provided some clarifications for each raised concern or comment (in bold) below and hope this is adequate in addressing them. We are happy to provide further clarification should this be needed.

  1. Abstract: The two sentences “The results showed that … up to 21%” and “Thus, it is recommended … weak subgrades” have no strong logic relationship with each other. The prior sentence indicates the benefits of thickness ≥250 mm, while this is not the complete reason that the thickness <250 mm “should not” be used. More explanation is required between the two sentences. (The sentence has been modified. Also, because this is the abstract, it is difficult to provide further explanation, however, modifying the sentence has removed the direct consequence relationship with the previous sentence, as suggested other factors play a role in the recommended thickness)
  2. Section 1: The sentence “Since pavement roughness and stiffness are influenced by climatic factors and the use of LCC is recommended due to its ability to mitigate some of these influencing factors like freeze-thaw cycling this study aims to address this gap by conducting post-construction evaluations of flexible pavement sections incorporating LCC as a subbase layer, with the objective of quantifying its benefits in terms of pavement, stiffness, and roughness” should be divided into two sentences. (This sentence has been divided into two)
  3. Section 3.1: The full names of AC, GA, and GB should also be explained in the title of Figure 3, so that readers would not check them in the main text. In addition, why there are two “(b) Erbsville LCC350 section view” in the title of Figure 3? This has been corrected and edited.
  4. The section numbers are not in sequence. For example, the subsection number of Section 2 is 3.1 (should be 2.1), two “section 3.1” in the paper, etc. The authors should be serious to these details. (This has been corrected)
  5. It is not very necessary to present Table 2 separately. The contents can be included in Figure 6 next to RWP and LWP. Same for Table 3 and Figure 8, Table 5 and Figure 11, Table 6 and Figure 12 (Tables and figures have been merged accordingly)
  6. Table 4: Other marks instead of red color should be used to highlight the results, considering the formatting and printing of the journal. (Red colour has been substituted with boxes)
  7. Figure 10: Axes names are missing. (Axes name has been added)
  8. Since LCC is advantageous for low carbon footprint, a short section with primary estimation of CO2-eq is required to complement the paper, considering the different raw materials production, construction, and service life between LCC and the control subbase (This was not the focus of this paper, however, this has already been done in another study see Oyeyi, Ni, and Tighe 2023).                                                       Abimbola Grace Oyeyi, Jessica Achebe, Frank Mi-Way Ni & Susan Tighe (2023) Life cycle assessment of lightweight cellular concrete subbase pavements in Canada, International Journal of Pavement Engineering, 24:1, DOI: 10.1080/10298436.2023.2168662
  9. Conclusion: There is a lack of outlook to figure out the future research based on the results (or limitations) of this paper. (This has been added in the last paragraph)

Reviewer 4 Report

Manuscript Title: In-Service Performance Evaluation of Flexible Pavement with Lightweight Cellular Concrete Subbase  by Abimbola Grace Oyeyi1,*, Frank Mi-Way Ni2 , and Susan Tighe 1,3*

Review Comments:

1.       It is a pleasure to review a well written technical manuscript that was obviously proofread prior to submission, nice job!

2.       Page 2, rows 67-72: The replacement of Portland cement 68 with fly ash, up to a maximum of 75%, in low-density lightweight concrete (LCC) can 69 decrease embodied carbon dioxide (eCO2) while also improving properties such as lower 70 thermal conductivity, reduced dry shrinkage, and diminished heat of hydration [14-17]. 71

How does the use of 75% influence constructability, in particular the time LCC must cure before heavy construction equipment can be operated on it to construct the remainder of the pavement structure?

3.       Page 4, rows 136-139: This study investigated the performance of flexible pavement sections incorporating 136 unbound granular materials as a subbase layer in the Control section, and three densities 137 of Lightweight Concrete (LCC) (400, 475, and 600 kg/m3 ) in the LCC sections. The in-ser- 138 vice evaluation and seasonal impact on pavement stiffness, and roughness were con- 139 ducted using Lightweight deflectometer and SurPro equipment. 140

Was any FWD testing done and correlated with LWD testing since the pavement structure thicknesses were significant?

4.       Page 4, Rows 157-160: The figure illustrates both test section project locations, but only one has been described. The figure should be move to below the first paragraph on Page 5 which describes the second test section project location.

5.       Pages 14-15 Roughness figures: The variability is so high in the measurements that coefficients of variation should be reported to help readers recognize the variation relative to the means.  

Author Response

Dear Reviewer,
Thank you very much for all the comments and suggestions. They were very helpful in updating this work and making sure that our work is well understood by the readers. We really appreciate the detail and effort put into reviewing this work.

We have provided some clarifications for each raised concern or comment (in bold) below and hope this is adequate in addressing them. We are happy to provide further clarification should this be needed.

  1. It is a pleasure to review a well written technical manuscript that was obviously proofread prior to submission, nice job!
  2. Page 2, rows 67-72: The replacement of Portland cement 68 with fly ash, up to a maximum of 75%, in low-density lightweight concrete (LCC) can 69 decrease embodied carbon dioxide (eCO2) while also improving properties such as lower 70 thermal conductivity, reduced dry shrinkage, and diminished heat of hydration [14-17]. 71

How does the use of 75% influence constructability, in particular the time LCC must cure before heavy construction equipment can be operated on it to construct the remainder of the pavement structure? In this, we only considered the use of 20% slag and this did not significantly affect the curing time compared with 100% cement content. However, the curing time was slightly faster (2 -4 hrs) for 100% cement LCC on the field. But it was noted that the strength development between 3 and 7 days was higher for LCC mixes with 20% slag. Also, the study cited with 75% showed there was no impact to strength of LCC when compared with 100% cement.

  1. Page 4, rows 136-139: This study investigated the performance of flexible pavement sections incorporating 136 unbound granular materials as a subbase layer in the Control section, and three densities 137 of Lightweight Concrete (LCC) (400, 475, and 600 kg/m3 ) in the LCC sections. The in-ser- 138 vice evaluation and seasonal impact on pavement stiffness, and roughness were con- 139 ducted using Lightweight deflectometer and SurPro equipment. 140

Was any FWD testing done and correlated with LWD testing since the pavement structure thicknesses were significant? FWD is currently being done and the results so far show a similar trend although greater Elastic modulus magnitude which is expected. Since this was not the focus of this paper, it has been included in the recommendation for further studies to perform and correlate LWD and FWD results.

4. Page 4, Rows 157-160: The figure illustrates both test section project locations, but only one has been described. The figure should be move to below the first paragraph on Page 5 which describes the second test section project location. (Figure has been moved down)

5. Pages 14-15 Roughness figures: The variability is so high in the measurements that coefficients of variation should be reported to help readers recognize the variation relative to the means.  (COV results have been included in the manuscript)